# Boron-Doped Diamond/GaN Heterojunction—The Influence of the Low-Temperature Deposition

**DOI:** 10.3390/ma14216328

**Published:** 2021-10-23

**Authors:** Michał Sobaszek, Marcin Gnyba, Sławomir Kulesza, Mirosław Bramowicz, Tomasz Klimczuk, Robert Bogdanowicz

**Affiliations:** 1Faculty of Electronics, Telecommunications and Informatics, Gdansk University of Technology, 11/12 Narutowicza Str., 80-233 Gdansk, Poland; marcin.gnyba@pg.edu.pl (M.G.); robbogda@pg.edu.pl (R.B.); 2Faculty of Technical Sciences, Warmia and Mazury University in Olsztyn, 11 Oczapowskiego Str., 10-719 Olsztyn, Poland; kulesza@matman.uwm.edu.pl (S.K.); miroslaw.bramowicz@uwm.edu.pl (M.B.); 3Department of Solid State Physics, Faculty of Applied Physics and Mathematics, Gdansk University of Technology, 80-233 Gdansk, Poland; tomasz.klimczuk@pg.edu.pl

**Keywords:** epitaxial gallium nitride, boron-doped diamond, heterojunction, AFM, electronic properties

## Abstract

We report a method of growing a boron-doped diamond film by plasma-assisted chemical vapour deposition utilizing a pre-treatment of GaN substrate to give a high density of nucleation. CVD diamond was deposited on GaN substrate grown epitaxially via the molecular-beam epitaxy process. To obtain a continuous diamond film with the presence of well-developed grains, the GaN substrates are exposed to hydrogen plasma prior to deposition. The diamond/GaN heterojunction was deposited in methane ratio, chamber pressure, temperature, and microwave power at 1%, 50 Torr, 500 °C, and 1100 W, respectively. Two samples with different doping were prepared 2000 ppm and 7000 [B/C] in the gas phase. SEM and AFM analyses revealed the presence of well-developed grains with an average size of 100 nm. The epitaxial GaN substrate-induced preferential formation of (111)-facetted diamond was revealed by AFM and XRD. After the deposition process, the signal of the GaN substrate is still visible in Raman spectroscopy (showing three main GaN bands located at 565, 640 and 735 cm^−1^) as well as in typical XRD patterns. Analysis of the current–voltage characteristics as a function of temperature yielded activation energy equal to 93.8 meV.

## 1. Introduction

Nowadays, single-crystalline gallium nitride (GaN) is widely known as the first-choice material for high-electron-mobility transistors (HEMTs) due to its high breakdown voltage and current handling capability. Unfortunately, this application is limited by the self-heating effect of GaN devices under operation [1]. Controlling the temperature in the working area of AlGaN/GaN HEMTs is crucial from the point of view of the reliability of such devices but also allows them to be optimised. However, SiC-based transistors do not exceed a direct current (DC) power density of 10 W/mm with a maximum junction temperature approaching 200 °C [2]. Therefore, the use of diamond might increase the power density dissipated in GaN or AlGaN/GaN HEMTs [3], due to the very high current density, high junction temperature, and high electric field [4].

GaN-on-diamond typically requires multiple steps of growth and transfers to produce structures [5]. On the contrary, the major issue of diamond on GaN is the lack of high-temperature stability of GaN under diamond chemical vapor deposition (CVD) process conditions. The typical substrate temperature for CVD diamond is around 800–900 °C, which may result in etching and decomposition of the GaN [6,7]. May et al. showed a multi-step process to achieve continuous diamond film on GaN without decomposition; however, the diamond quality is poor and has a cauliflower morphology [7].

Currently, the CVD of diamond on GaN substrates is carried out with a transition layer such as AlGaN or Si_3_N_4_. The latest work by Babchenko et al. [8] shows a high electron mobility transistor fabricated from diamond/SiNx-coated AlGaN/GaN. The diamond layer was deposited by a low-temperature microwave plasma assisted chemical vapor deposition (LAMWPCVD) system on the SiN_x_ interlayer, resulting in a rich sp^2^ diamond film. Next, Malakoutian et al. integrated polycrystalline diamond with N-polar GaN by the PACVD process. The authors used a 5 nm thin Si_3_N_4_ dielectric layer to reduced H_2_ plasma etch the GaN HEMT [9]. A SiN_x_ interlayer was used by Ahmed et al. [4]. To summarise, there is still space for studies on polycrystalline diamond deposition directly on GaN substrates or devices, reducing the complexity of the technological processes, and so on.

Here, we report the results of the direct deposition of polycrystalline boron-doped diamond on epitaxial GaN (BDD@GaN) without a transition layer by the low-temperature microwave plasma-assisted chemical vapour deposition process. The GaN substrates were pre-treated with hydrogen plasma to achieve a higher density of nucleation. This results in an expansion of the surface of GaN substrate; the irregular topography of the substrate significantly increases the adhesion of the diamond layer to the GaN surface. This is critical due to cooling cracks or small gaps opening up at the diamond/GaN interface, resulting in partial or full delamination and negligible thermal conduction through the interface, which was shown by Smith et al. [10]. In view of the fact that the lack of a covalent bond between Ga-C can lead to delamination of the diamond layer, solutions that increase the adhesion of the layer to GaN are very desirable. It is also worth noting that additional internal stress resulting from the PACVD process was not observed in the produced BDD@GaN heterojunction, which was confirmed in the Raman spectroscopy.

Furthermore, the samples also were investigated by scanning electron microscopy (SEM), atomic force microscopy (AFM), and X-ray diffraction spectroscopy (XRD). Additionally, the electrical properties of the structures were investigated, aiming at their possible application in semiconductor devices.

## 2. Materials and Methods

### 2.1. CVD Growth

The substrate for the boron-doped diamond was gallium nitride (GaN) grown epitaxially via the molecular-beam epitaxy process on silicon [11]. Prior to the deposition of the diamond films, a pre-treatment procedure was applied to the e:GaN substrates. The substrates were cleaned in acetone and isopropanol in an ultrasonic bath and subsequently hydrogen plasma-treated for 5 min. The BDD/GaN heterojunctions were synthesized in an MW PA CVD system (Seki Technotron AX5200S, Tokyo, Japan). The heated stage was set to 500 °C and the microwave power was limited to 1100 W. Next, the pre-treated substrates were sonicated in a water-based nanodiamond suspension containing diamond particles 4–5 nm in size (Adamas Nanotechnologies, Nagoya, Japan). An H_2_/CH_4_ gas mixture of 1% vol. with an overall gas flow of 300 sccm and temperature of 500 °C was used. The films presented in the results section were grown at a pressure of 50 Torr and PMW = 1100 W of microwave power, resulting in a high-density direct plasma at the region of the substrate. Diborane (AirLiquide, Kraków, Poland) (B_2_H_6_) was used as a boron dopant precursor. The boron doping level expressed as the [B]/[C] ratio in the gas phase was set to 2k or 7k ppm. The growth time of the BDD films was 2 h, producing sub-microcrystalline films with an average thickness of 442 for 2k and 483 nm for 7k film.

### 2.2. Microscopic Imaging

A scanning electron microscope (S-3400N, Hitachi, Japan) with a tungsten source and variable chamber pressure scanning electron microscope (VP-SEM) was utilized to investigate the surface of the synthesized thin films.

### 2.3. Nanoscale Imaging

Specific surface properties of the B-doped polycrystalline diamond films on GaN epilayers were examined using an AFM (Multimode 8, Bruker, Billerica, MA, USA) working in the specific tapping mode called PeakForce Tapping that enables direct control of the tip-sample force at the level of piconewtons. In this mode, the tip periodically approaches the surface with the sub-resonant frequency and monitoring maximum contact force set at 500 pN, while the probe continues the raster scan. Real-time data is processed on the fly in order to determine the local topography and mechanical properties of the surface, which is known as quantitative nanomechanical mapping (QNM). Each image consists of samples taken from a 2 × 2 µm^2^ scan area probed in 256 × 256 steps (the lateral resolution is 7.81 nm). The scanning probes were made of silicon and had a nominal spring constant of 5 N/m, a resonance frequency of 160 kHz, and a tip radius of 8 nm (HQ:NSC14/Al-BS, Mikro-Masch, Tallinn, Estonia).

### 2.4. Raman Spectroscopic Setup

The molecular composition of the films was studied by means of Raman spectroscopy using a Raman confocal microscope (Horiba LabRAM ARAMIS, Tokyo, Japan). Spectra were recorded in the range of 200–3500 cm−1 with an integration time of 2 s, using a 532 nm diode-pumped solid-state (DPSS) laser in combination with a 100× objective magnification (NA = 0.95) and 50 μm confocal aperture.

### 2.5. XRD Setup

The x-ray diffraction measurement was performed on a plate-like sample using a Bruker D2Phaser diffractometer with CuKα radiation, equipped with a high-speed, high-resolution XE-T detector.

### 2.6. Electrical Properties

The resistance of the BDD@GaN was measured in the temperature range of 293–573 K by a two-point probe. The needles with a diameter of 20 μm were connected to the measuring source unit (Keithley 2400, Tektronix, Oldbury, Bracknell, UK) via two micro-positioners (Signature S-725), which were mounted on the x-y stage.

## 3. Results and Discussion

### 3.1. Micro and Nanoscopic Investigations of BDD Surfaces Grown on Epitaxial GaN

The GaN substrates exhibited a negative value of ζ potential, due to absorbed oxygen on the surface [12]. This characteristic strengthens on exposure to hydrogen plasma, due to the change of oxygen to hydroxyl, which undergoes protolysis and adsorption on the surface [13]. Next, they were treated with H-terminated seeds, which have a positive ζ potential, resulting in a high seeding density [12] and a good-quality diamond film. Figure 1 shows a cross-section of the BDD film, where the interface between the BDD film and the GaN is clearly visible. The average thickness of the BDD film after two hours of deposition was 480 nm (431 nm for 2k sample).

The surface morphology of the prepared samples was investigated by AFM. Figure 2 shows nanoscale images of the height variations and mechanical properties of the B-doped polycrystalline films on GaN epilayers taken using the AFM method in PF-QNM tapping mode with a scan size of 2 × 2 µm^2^. Data processing yielded measures of spatial inhomogeneities that are summarised in Table 1. The AFM images shown in Figure 2 exhibit surface morphologies of the BDD samples common for polycrystalline, closed films with randomly aligned crystallites without any predominant surface texture. The observed crystal habits defined by the specific shape of their facets (either triangular or rectangular) and sharp edges confirm the presence of well-developed polycrystals that took control of the growth space during the nucleation step but are spatially confined so that they can further expand only within the competitive growth regime.

AFM is widely used in the characterisation of the surface properties of solids. In connection with numerical methods of image analysis, it provides a comprehensive insight into various aspects of surface phenomena. AFM investigations of polycrystalline diamond films have demonstrated, among others, irregular evolution of the surface roughness of the growing crystals [14], delimiting subsequent growth modes of the CVD process [15], and that increasing the level of boron dopants flattens the surface of the films [16]. On the other hand, interest in using AFM methods in studies on GaN/diamond sandwich structures appears to be focused on simple tasks, among others: basic characterisation of the surface roughness upon modification steps [17], validation of the overall crystal quality [18] and the required surface smoothness [19], investigation of structural damage within the films caused by additional coating procedures [20], etc.

Although nearly identical in terms of morphological attributes, the crystals in the low-doped sample (2000 ppm) exhibit mostly octahedral habits with characteristic (111) facets without traces of secondary nucleation. On the other hand, the structure of the highly doped sample (7000 ppm) shows a mixture of octahedral (111) facets with some amount of rectangular (100) ones. However, the number of newly grown diamond seeds is significantly higher than in the previous sample, which suggests a link between secondary nucleation and intense boron doping. The maps of Young’s pseudo-modulus and adhesion forces shown in Figure 2 reveal that the films under study are of good quality despite their polycrystalline and thus discontinuous structure, because their properties do not vary among grains, and neither phase nor chemical contaminations can be seen on the respective AFM images excluding disturbed regions, that is, grain boundaries and interfacial layers. The significantly higher adhesion forces seen at the grain boundaries compared to the bulk material are likely due to the presence of accumulating capillary layers in the basins between grains.

Table 1 summarises the spatial characteristics of the surface height variations which generally agree with the aforementioned visual observations of the specific surface patterns shown in the AFM images in Figure 2. The estimated planar grain concentration was 35 µm^−2^, regardless of the doping level; however, the BDD-2k@GaN was found to be the smoother among the two considering its lower surface roughness S_q_, despite the absence of the waviness removed during image preprocessing. On the other hand, the ISAD (relative difference between the surface area triangulated by the AFM height samples and a projected plane) was found to be lower in the high-doped sample. The above results appear counter-intuitive but might be really considering the aforementioned balance of various crystal habits in the films. The low S_q_ together with the large ISAD values correspond to a smooth surface with inclined crystal facets ((111) plane) as in the BDD-2k@GaN. The opposite relation (providing the same grain concentration) means a rough surface with sharp, vertical facets ((100) plane). Since the observed differences are small, the same concerns the relative contributions of both planes. Additionally, both samples were found to be isotropic regardless of the doping level, as the anisotropy ratio S_tr_ was found to be ca. 0.8, which confirms the independence of the polycrystalline alignment of the direction of observation, and thus the random orientation of the grains in the films.

The mean grain diameters D_AC_ estimated using the method of the autocorrelation decay described elsewhere [21] were 100 and 130 nm for the 2000 and 7000 ppm samples, respectively. On the other hand, Otsu’s thresholding method revealed much smaller grains (75–80 nm). Regardless of the method, however, the grain size was found to be at least an order of magnitude larger than the lateral resolution of the AFM scans, proving a sufficient image resolution. Previous work [22] demonstrated that the autocorrelation method relying on Full Width at Half Maximum (FWHM) values of the main peak overestimates the grain size with respect to other methods. Note also that the apparent size of the crystals measured in the AFM image is lower than that seen in the cross-section due to columnar expansion of the film towards the top of the sample.

Unlike the spatial parameters, nanomechanical characterization revealed clearer differences between the discussed films. The BDD-7k@GaN was found to be around 50% stiffer in terms of a larger Young’s pseudo-modulus possibly due to saturation of vacancies and site defects by lighter B-dopants, which strengthens the overall structure. Additionally, the mean adhesion forces were larger in this sample, however, the dependence between the doping level and tip–surface interaction is not clear at the moment due to perturbations introduced by the intermediate capillary layer.

### 3.2. Molecular and Crystallographic Studies of BDD Films Deposited on Epitaxial GaN

The molecular composition and quality of polycrystalline diamond films, GaN films, and crystals as well as Si substrates can be efficiently investigated by means of Raman microscopy. The Raman spectra of polycrystalline diamond and boron-doped diamond BDD have already been investigated by other groups, e.g., Ferrari et al. [23], and by our group [24]. The integral intensity, width, and position of the Raman band assigned to the diamond lattice (1332 cm^−1^) can be used to study the content of the crystalline sp^3^ phase, while the D band and G band can be used to determine the content of a distorted or amorphous sp^2^ phase, which decreases the quality of the diamond thin films. Moreover, the shift and asymmetry of the band assigned to the diamond lattice can be used to determine the stress of the lattice and the content of dopants, e.g., boron. The Raman spectra of a GaN layer have been presented in detail by, e.g., Siegle et al. [25] and Kaschner et al. [26], who studied crystalline hexagonal and cubic GaN deposited on sapphire. They showed the frequencies and symmetries of the modes found in the Raman spectra of GaN and their assignments, including three strong bands at 569 cm^−1^ (E_2_ mode), 640 cm^−1^ (A_1_ overtone), and 735 cm^−1^ (E_1_ mode), which are typical for GaN. Assuming transparency of the GaN and BDD layers, it confirms that Raman microscopy can be used to determine the quality and crystallinity of the investigated multi-layer structure.

The Raman spectra of the BDD films deposited on the GaN substrate are shown in Figure 3. The spectra were compared with crystalline GaN and BDD deposited on silicon. The decomposed Raman spectra of the BDD samples are shown in Figure 4. Analysis of the spectra confirms that the polycrystalline diamond layers were deposited directly on the surface of the GaN. The bands at 565 cm^−1^, 640 cm^−1^, and 735 cm^−1^ are assigned to GaN oscillation modes E_2_, A_1_, and E_1_, respectively [25,26]. Their presence confirms that the GaN crystalline network was not distorted during the CVD deposition of the diamond. The wideband at 970 cm^−1^ is due to the silicon substrate. The band at 1130 cm^−1^ can be assigned to nanocrystalline diamond structures (NCD) and the strong band at about 1330–1332 cm^−1^ comes from the oscillation mode of the diamond lattice. The bands at 1470 cm^−1^ and 1560 cm^−1^ can be assigned to non-diamond carbon-based structures of the deposited films trans-polyacetylene (TPA) and amorphous sp^2^ carbon (“G” band), respectively. The parameters of the Raman band are shown in Table 2, including the intensity ratio of the bands assigned to the diamond lattice and the “G” band, are in good agreement with the Raman spectra of the most common polycrystalline diamond films deposited, e.g., on silicone, which has already been shown in the literature [23,24]. The slight content of non-diamond carbon components (bands assigned to them are strong for green laser excitation) is typical for polycrystalline diamonds. The efficiency of the boron doping is confirmed by the slight shift of the Raman band assigned to the diamond lattice. Its central wavenumber shift moves slightly from 1332 cm^−1^, which is typical for undoped diamond, through 1331 cm^−1^ for the sample having the doping level equal to 2000 ppm, to 1330 cm^−1^ recorded for the doping level of 7000 ppm. Moreover, this band is widened from 13.23 cm^−1^ (FWHM) in the case of the 2000 ppm doping level to 16.26 cm^−1^ (FWHM) in the case of the 7000 ppm, which refers to distortions of the diamond lattice caused by boron atoms. Moreover, the presence of the band assigned to the silicon substrate in the Raman spectra confirms the transparency of the GaN layers and the BDD layers, which also confirms the low content of non-diamond carbon components and thus the high quality of the BDD layers. The molecular composition of the deposited structures confirms the deposition of a crystalline diamond layer, not an amorphous carbon layer directly on the surface of the e:GaN, which is necessary to ensure high thermal conductivity.

Raman spectroscopy can also reveal induced stress in the GaN film. The final stress in the GaN depends on the conditions of the diamond deposition process, such as the temperature, which induces internal stress in the diamond but can also induce it in the substrate [17]. As the E_2_ phonon frequency was equal to 565 cm^−1^ and 564 cm^−1^ for BDD-2k@GaN and BDD-7k@GaN, the deposition process of the BDD film did not induce stress in the GaN substrate, with reference to the relaxed GaN E_2_ phonon frequency value of 567.2 cm^−^^1^ [27].

Figure 5 presents the X-ray diffraction pattern with the intensity on the square root scale plotted versus the 2Θ angle. Several very intense Bragg reflections can be observed between 31 and 37 degrees. The d-lattice spacing for the first and the last reflection are d = 2.86 Å and 2.48 Å, respectively. A typical *c* lattice parameter for GaN (wurtzite form) is 5.185 Å. However, Petkov et al. studied the crystal structure of the nanocrystalline form of GaN and obtained a 10% larger value of *c* = 5.73(2) Å [28]. This strongly suggests that the first seen reflection is (002) and the one near 2Θ = 65 deg. is (004) of a GaN layer. It is worth noting that the GaN layer orientation is not perfect, as proved by observation of a (110) reflection with d = 1.57 Å, which gives a = 3.14 Å—again in perfect agreement with a value reported in [28].

The size of crystallites was calculated from XDR spectra by Scherrer Equation (1):(1)$$D=\frac{K\lambda \;}{\beta cos\theta }$$
where *D* is the mean size of the crystallites, *K* is a dimensionless shape factor 0.94, *λ* is the X-ray wavelength CuKα = 1.54 Å, *β* is the line broadening at half the maximum intensity (FWHM), and *θ* is the Bragg angle. The calculated size is equal to 29.9 nm, so the crystallites are smaller compared to grains with an average size of 100 nm.

### 3.3. Electronic Properties of BDD Films Grown on Epitaxial GaN Surfaces

The resistance values recorded between the contact top and bottom of the BDD-7k@GaN followed a linear distribution, while the measured values of R were 16 kΩ and 3.6 kΩ at room temperature and T = 520 K, respectively. Figure 6 shows measured conductance G versus 1/T, where G was extracted from each trace and expressed as a natural logarithm. The inverse conductance data followed an approximately linear distribution from T = 373 K values to T = 523 K (see inset of Figure 6). In the case of temperature T = 373 K, the distribution is much less linear, which may be due to defects formed during the synthesis of the BDD. The slope analysis of the activation plot from the diamond-on-GaN heterostructure yielded an energy barrier of E_a_ = 93.8 meV from the fitted slope E_a_/k_B_ = 1088.88 (see Figure 6). This heterojunction exhibited a typical semiconducting behaviour in the Arrhenius plot with no thermally activated conductance.

The obtained activation energy is similar to reports in the literature for BDD films [29,30]. It is worth noticing that the heterojunction of boron-doped diamond on GaN substrates has a significantly lower E_a_ than other semiconductors or semiconductor devices such as SiC Metal-Oxide Semiconductor Field-Effect Transistors (MOSFETs) and AlGaN/GaN (see Table 3).

## 4. Conclusions

We have reported the direct deposition of boron-doped diamond films onto epitaxially grown GaN using plasma-assisted chemical vapor deposition. This allows to achieve a diamond/GaN interface free of an intermediate or interfacial layer and, thus, in direct contact. Suitable conditions for successful PACVD growth were achieved through applying an exposure of the GaN to hydrogen plasma as pre-treatment prior to nucleation. The SEM analysis revealed full encapsulation of GaN substrates and a multi-faceted polycrystalline morphology. AFM measurements in PF-QNM mode revealed that well-developed diamond grains are on average 100 nm in diameter, tightly joined to form closed but isotropic films. The calculated crystallite size is noticeably smaller than grain reaching 29.9 nm. Both samples do not exhibit internal stress, after the PACVD process, which was confirmed by Raman spectroscopy, which showed E_2_ phonon frequency was equal to 565 cm^−1^ and 564 cm^−1^ for the BDD-2k@GaN and BDD-7k@GaN in comparison to relaxed GaN E_2_ value 567.2 cm^−1^. Next, the mechanical stiffness and surface adhesion were found to be significantly larger in the BDD-7k@GaN than BDD-2k@GaN, resulting in Y_mod_ 1280 ± 640 and adhesion force 2.84 ± 0.87.

Finally, the electrical measurements showed that the activation energy of this BDD-7k@GaN heterojunction was equal to 93.8 meV, which is significantly lower than for, e.g., SiC-based devices. However, despite the technical difficulties in coating GaN with diamond films, searching for solutions enabling the deposition of diamond layers directly on GaN is highly demanded.

## Figures and Tables

**Figure 1 materials-14-06328-f001:**
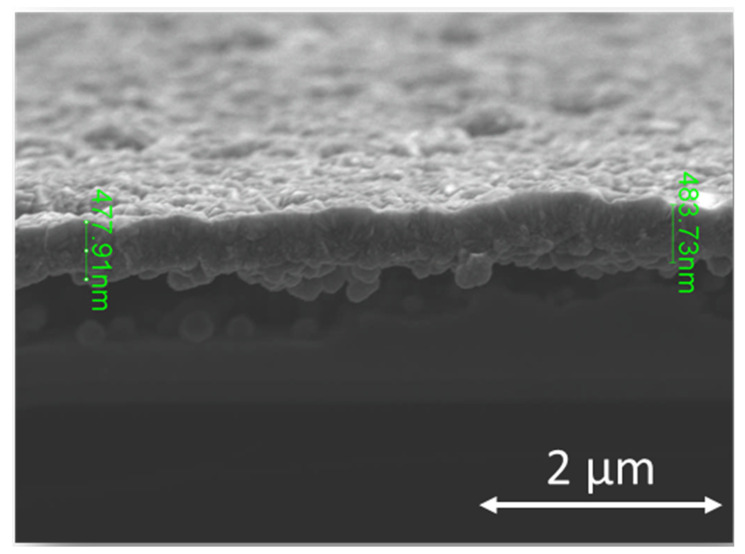
SEM cross-section of boron-doped diamond film deposited on e:GaN with B/C ratio 7000 ppm.

**Figure 2 materials-14-06328-f002:**
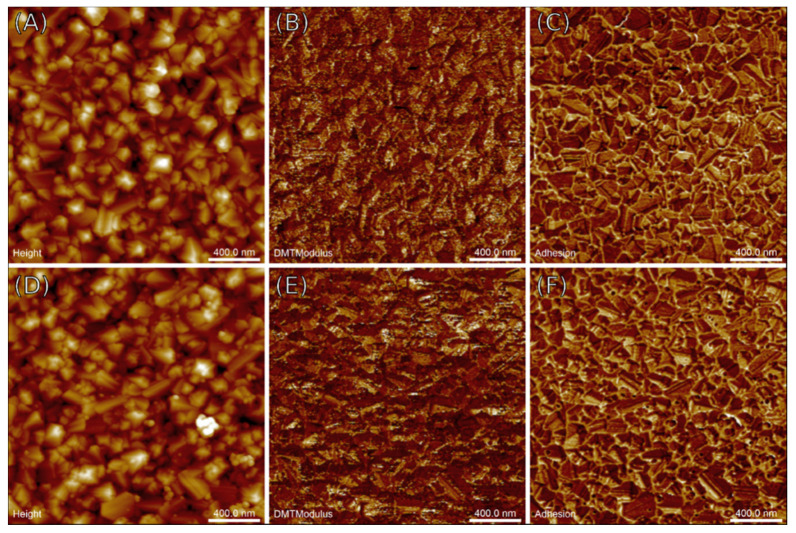
AFM images of the BDD films with different boron doping levels: (**A**–**C**) 2000 ppm and (**D**–**F**) 7000 ppm made in PF-QNM mode and subsequently separated into data channels revealing various aspects of the surface inhomogeneities: (**A**,**D**) surface topography (height), (**B**,**E**) Young’s pseudo-modulus (DMT modulus), and (**C**,**F**) tip-surface adhesion force (adhesion).

**Figure 3 materials-14-06328-f003:**
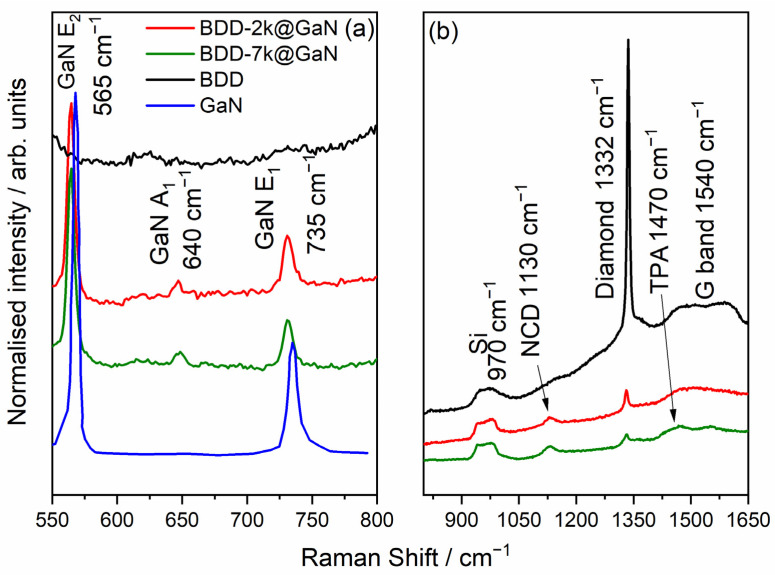
Raman spectra of boron-doped diamond films with different boron doping levels: red curve 2000 ppm, green curve 7000 ppm. (**a**) Raman spectra form 550 cm^−1^ to 800 (**b**) Raman spectra form 875 cm^−1^ to 1650.

**Figure 4 materials-14-06328-f004:**
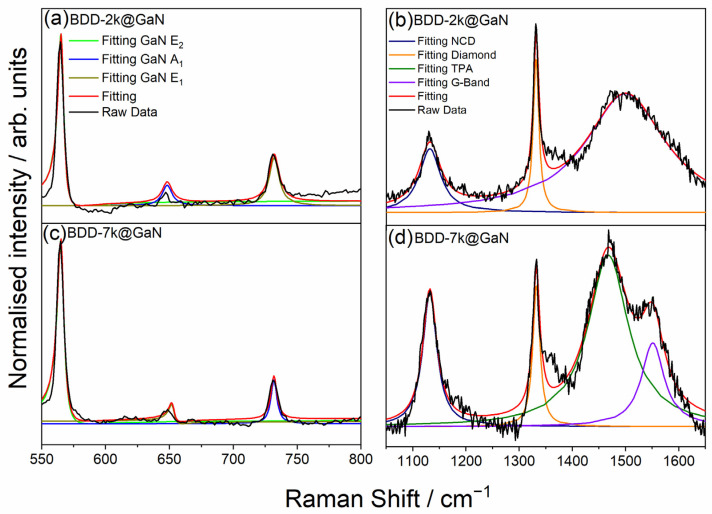
Normalised Raman spectra modelled using the Breit-Wigner-Fano function: (**a**,**b**) BDD2k and (**c**,**d**) BDD7k.

**Figure 5 materials-14-06328-f005:**
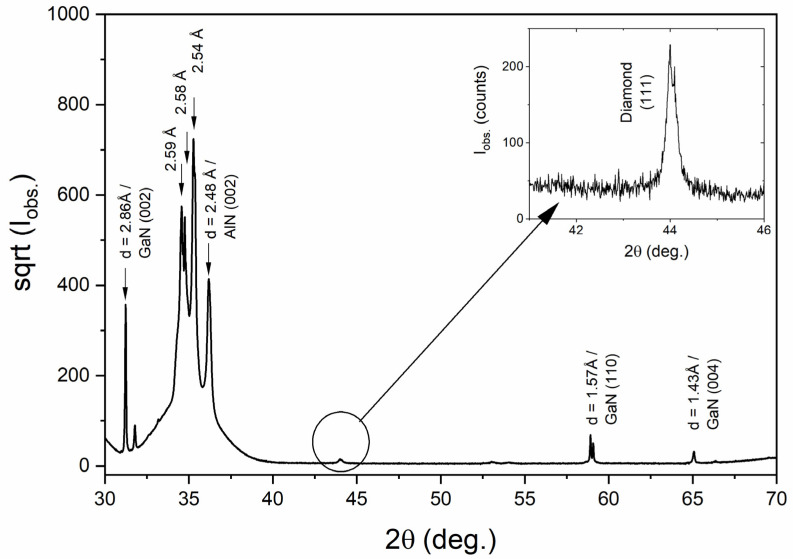
X-ray diffraction pattern for BBD-7k@GaN. The intensity is on square root scale. The inset shows a small (111) reflection of a diamond phase.

**Figure 6 materials-14-06328-f006:**
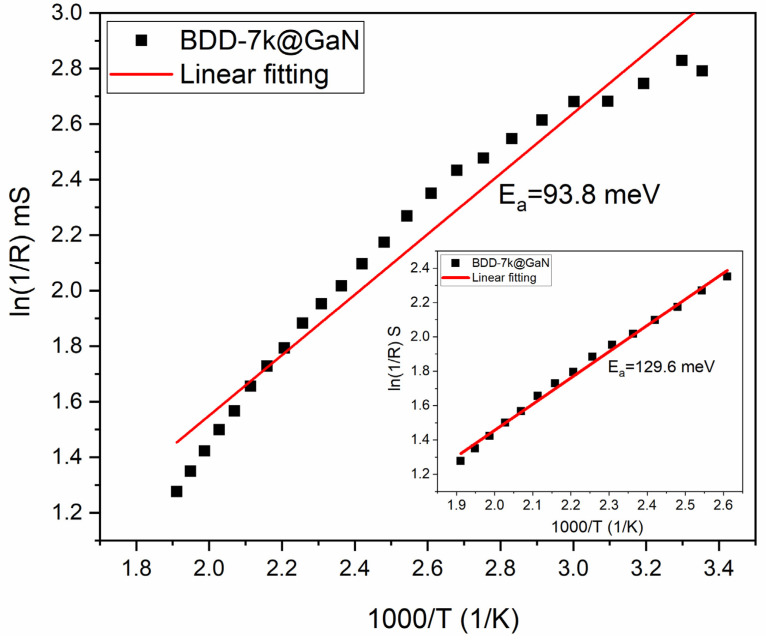
Arrhenius plot for BDD-7k@GaN sample with boron doping 7000 ppm B/C.

**Table 1 materials-14-06328-t001:** Surface texture and mechanical characteristics derived from AFM images: S_gr_—planar grain concentration, S_q_—surface roughness (RMS), ISAD—image surface area difference, S_tr_—texture anisotropy ratio, D_AC_—mean grain size from the autocorrelation function, D_Otsu_—grain size estimated using Otsu’s method, Y_mod_—mean Young’s pseudo-modulus, F_adh_—mean adhesion force.

Sample	S_gr_ [µm^−2^]	S_q_ [nm]	ISAD [%]	S_tr_	D_AC_ [nm]	D_Otsu_ [nm]	Y_mod_ [MPa]	F_adh_ [nN]
BDD-2k@GaN	35 ± 2	12.7 ± 1.2	9.86 ± 0.32	0.80 ± 0.04	100 ± 15	80 ± 2	877 ± 351	1.51 ± 0.32
BDD-7k@GaN	35 ± 3	13.6 ± 1.4	8.78 ± 0.54	0.81 ± 0.07	130 ± 18	75 ± 4	1280 ± 640	2.84 ± 0.87

**Table 2 materials-14-06328-t002:** The parameters of the Raman bands for BDD-2k@GaN and BDD-7k@GaN.

Sample	Peak	Peak Position	Norm. Intensity	FWHM
BDD-2k@GaN	GaN E_2_	565	0.05032	6
	GaN A_1_	648	0.00597	9.48
	GaN A_2_	731	0.01396	9.54
	NCD	1131	0.01076	48.44
	Diamond	1331	0.02584	13.23
	G-Band	1540	0.0201	183.07
BDD-7k@GaN	GaN E_2_	564	0.04951	6.76
	GaN A_1_	648	0.00514	6.86
	GaN A_2_	731	0.01174	5.70
	NCD	1131	0.01151	32.25
	Diamond	1130	0.01207	16.26
	TPA	1467	0.0147	91.77
	G-Band	1550	0.00717	51.94

**Table 3 materials-14-06328-t003:** Comparison of the activation energy for different semiconductor devices.

Semiconductor Material	Process	Activation Energy	Reference
Phosphorous-doped diamond	MPCVD	0.54 eV	[31]
Homoepitaxial boron-doped diamond films	MPCVD	185 meV	[29]
BDD	MPACVD	314 meV to 101 meV	[30]
6H-SiC	VPE	0.13 eV	[32]
AlGaN/GaN HFET @ SiC	nd	0.38 eV	[33]
SiC MOSFET	nd	0.9 eV	[34]
SiC MOSFET	nd	1.1 eV	[35]
BDD-7k@GaN	MPACVD	93.8 meV	this work

## Data Availability

Not applicable.

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
