# Peer review of "Boron-Doped Diamond/GaN Heterojunction—The Influence of the Low-Temperature Deposition"

_materials, 2021, doi:10.3390/ma14216328_

Round 1

Reviewer 1 Report

Authors reported that Polycrystalline boron-doped diamond films are directly deposited onto epitaxial gallium nitride substrates by MPCVD method. and the electrical properties of the structures were investigated aiming at their possible application in semiconductor devices, which is beneficial for the application of board bandgap semiconductor in electronics device. However, some questions should be answered befor accepted.

  1. Figue2 captions should be descripted in details, including what's mean of a-f?
  2. what happend to the substrate of GaN under the microwave plasma? how about the damage?
  3. what 's the epitaxial mechanism BBD on the GaN?

Reviewer 2 Report

The manuscript is interesting and novel and is devoted to deposition of polycrystalline boron-doped diamond films. However, it should be improved as some data or text is missing or results not well presented.

  1. Paragraph presented in Line 90-95 should be removed or re-written.
  2. The authors should indicate the main aim of the work at the end of introduction section. The introduction should be focused on the novelty of the work, but not to Raman results description of GaN films.
  3. The XRD diffraction figure is not given.
  4. The error bars should be included in the Table1.
  5. The information presented in Lines 62-89 is more suitable for explanation of the obtained results.
  6. Line 11: producing sub-microcrystalline films with an average thickness of 442 for 2k and 483 nm for 7k film. SEM image (Fig.1) of diamond film clearly indicate that the thickness is slight different. Thus please add the error bars and give average values.
  7. The notes in figure 4 are very small.
  8. The G peak, TPA peak positions and FWHM values obtained from Raman spectra (Fig.4 b and d) should be presented in the text or figure.
  9. Line 318: The SEM analysis revealed full encapsulation of GaN substrates and well-formed crystallites. The SEM does not give information about crystallinity of the films.
  10. The conclusions should be re-formulated. Please indicate the difference between the deposited diamond films.
  11. The authors should very carefully revised the text of manuscript, because there are many inaccuracies in the work. Also the abstract should be improved.
  12. The size of crystallites should be calculated from XRD data.
